Hematopoietic effect of echinochrome on phenylhydrazine-induced hemolytic anemia in rats

El-Shehry Mona S. E. F. 1
Amrymi Rafa A. 2
http://orcid.org/0000-0001-8236-5051 Atia Tarek 3
Lotfy Bassant M. M. 1
Ahmed Salma H. A. 1
Qutb Sarah A. 4
Ali Sara B. 4
Mohamed Ayman S. 4
Mousa Mohamed R. 5
Damanhory Ahmed A. 6 7
Metawee Mostafa E. 8 9
http://orcid.org/0000-0003-2917-2423 Sakr Hader I. 10 11 hadersakr@kasralainy.edu.eg
1 Biotechnology Department, Faculty of Biotechnology-October University for Modern Sciences and Arts (MSA) , Cairo , Egypt
2 Department of Zoology, Faculty of Arts and Sciences, Alabyar University of Benghazi , Benghazi , Libya
3 Department of Medical Laboratory, College of Applied Medical Sciences, Prince Sattam bin Abdulaziz University , Al-Kharj , Saudi Arabia
4 Zoology Department, Faculty of Science—Cairo University , Cairo , Egypt
5 Department of Pathology, Faculty of Veterinary Medicine, Cairo University , Giza , Egypt
6 Department of Biochemistry, General Medicine Practice Program, Batterjee Medical College , Jeddah , Saudi Arabia
7 Department of Biochemistry, Faculty of Medicine, Al-Azhar University , Cairo , Egypt
8 Department of Histology, Faculty of Medicine, Al-Azhar University , Cairo , Egypt
9 Department of Histology, General Medicine Practice Program, Batterjee Medical College , Jeddah , Saudi Arabia
10 Department of Medical Physiology, Faculty of Medicine, Cairo University , Cairo , Egypt
11 Department of Medical Physiology, General Medicine Practice Program, Batterjee Medical College , Jeddah , Saudi Arabia
Siddiqui Arif
Electronic publication date: 2023 Dec 8
Publication date: 2023
Volume: 11
Electronic Location ID: e16576
Received 2023 Apr 6; Accepted 2023 Nov 13
Copyright: © 2023 El-Shehry et al.
Copyright year: 2023
Copyright holder: El-Shehry et al.
License: This is an open access article distributed under the terms of the Creative Commons Attribution License, which permits unrestricted use, distribution, reproduction and adaptation in any medium and for any purpose provided that it is properly attributed. For attribution, the original author(s), title, publication source (PeerJ) and either DOI or URL of the article must be cited.
License URL: https://creativecommons.org/licenses/by/4.0/

Keywords: Hemolytic Anemia, Echinochrome, Oxidative stress, Phenylhydrazine

Funding: The authors received no funding for this work.

==============================
Background

Hemolytic anemia (HA) is a serious health condition resulting from reduced erythrocytes’ average life span. Echinochrome (Ech) is a dark-red pigment found in shells and spines of sea urchins.

Aim

Studying the potential therapeutic effect of Ech on phenylhydrazine (PHZ)-induced HA in rats.

Methods

Eighteen rats were divided into three groups (n = 6): the control group, the phenylhydrazine-induced HA group and the Ech group, injected intraperitoneally with PHZ and supplemented with oral Ech daily for 6 days.

Results

Ech resulted in a considerable increase in RBCs, WBCs, and platelets counts, hemoglobin, reduced glutathione, catalase, and glutathione-S-transferase levels, and a significant decrease in aspartate & alanine aminotransferases, alkaline phosphatase, gamma-glutamyl transferase, bilirubin, creatinine, urea, urate, malondialdehyde & nitric oxide levels in anemic rats. Histopathological examination of liver and kidney tissue samples showed marked improvement.

Conclusion

Ech ameliorated phenylhydrazine-induced HA with a hepatorenal protective effect owing to its anti-inflammatory and antioxidant properties.

Introduction

The World Health Organization (WHO) estimates that one-fourth of the world’s population suffers from anemia, with almost half of the cases occurring in preschool-age children (Gallagher, 2022). According to the WHO, anemia is defined as low hemoglobin (Hb) level below 12.0 g/dL in women or <13.0 g/dL in men (Milovanovic et al., 2022). The reduced oxygen-carrying capacity of RBCs in anemic patients manifests clinically as lethargy, weakness, dizziness, blackouts, light headedness, shortness of breath, or even arrhythmias (Baldwin, Pandey & Olarewaju, 2022).

Based on the mean corpuscular volume (MCV) of RBCs, anemia is classified into micro-, normo-, or macrocytic. Hemolytic anemia (HA) is considered a subtype of normocytic anemia (MCV = 80 to 100 fL), with low Hb concentration due to the excess or premature RBCs destruction, secondary to Baldwin, Pandey & Olarewaju (2022). Intracorpuscular or extracorpuscular causes. Most intracorpuscular HAs are inherited disorders, however, extracorpuscular HAs are secondary to various exogenous factors including toxins, infections, and auto- or alloantibody against RBCs membrane antigens (Voulgaridou & Kalfa, 2021).

Low iron diets and deficiencies in vitamin B12 and folic acid are animal models for anemia. Due to the long induction period required by these methods, the anti-cancer drug cyclophosphamide or phenylhydrazine (PHZ) are typically used (Lee et al., 2014). PHZ is a strong oxidant that yields reactive oxygen species (ROS), phenyldiazene, phenylhydrazyl radical, and benzene diazonium ions. PHZ metabolites damage and cause severe HA by increasing levels of oxy-hemoglobin. PHZ interacts with membrane cytoskeleton, which causes spectrin degradation and RBCs deformation. PHZ also causes lipid peroxidation, cation and cytoskeletal imbalances, and reduces membrane deformability (Moreau et al., 2012). PHZ oxidizes oxyhemoglobin to methemoglobin, then to irreversible hemichromes, and finally precipitates forming Heinz bodies (Soliman, Mohamed & Marie, 2016).

Sea urchins (Paracentrotus lividus) are well-known species that inhabit the Atlantic and Mediterranean coasts. They are small, spiny, and globular animals belonging to the echinoderm phylum’s class “Echinoidea.” According to their primary sources, they contain polyhydroxylated naphthoquinone (PHNQ) pigments, either echinochromes or spinochromes. Moreover, sea urchin shells have shown to have some medical uses as pharmaceutical antioxidants (Soliman, Mohamed & Marie, 2016). PHNQ pigments have considerable antimicrobial and antioxidant activity. The water-insoluble echinochrome (Ech) is one of multiple spinochrome pigments found in the shells, spines, eggs, and gonads of sea urchins (Sayed, Soliman & Fahmy, 2018). Ech exhibits in vitro antiviral action against against RNA-containing tick-borne encephalitis virus and DNA-containing herpes simplex virus type 1 (Fedoreyev et al., 2018). Additionally, Ech shown in vitro antibacterial efficacy against Salmonella typhimurium, Pseudomonas aeroginosa, Staphylococcus aureus, and Listeria monocytogenes (Sadek et al., 2022). It mediates cellular responses, scavenges radicals, activates glutathione, increases mitochondrial function, reduces ROS imbalance, and inhibits lipid peroxidation (Rubilar et al., 2021). Ech is used as an active ingredient in the histochrome drug (Jeong et al., 2014). This drug is already being used commercially in Russia to cardiac diseases (Hwang et al., 2021) and phthalmological disorders (Egorov et al., 1999). Furthermore, echinochrome A has a physiological impact similar to ascorbic acid (Ageenko, Kiselev & Odintsova, 2022). Furthermore, Ech’s therapeutic activities in asthma management involve regulating the Keap1/Nrf2 signaling pathway to reduce inflammation and oxidative stress (Abdelmawgood et al., 2023). The principal aim of this study is to investigate the efficacy of Ech pigment against phenylhydrazine-induced HA in rats.

Materials and Methods

Chemicals and reagent

Dimethyl sulfoxide (DMSO) and phenylhydrazine (PHZ) were obtained from Sigma-Aldrich (St. Louis, MO, USA). All kits were purchased from the Biodiagnostic Company (El Motor St, Dokki, Egypt). Standard Ech was produced by G.B. Elyakov Pacific Institute of Bioorganic Chemistry (Vladivostok, Russia).

Sea urchin collection

Sea urchins were collected from Alexandria City’s coast (Egypt), thoroughly washed to remove sands and other organisms, packed in ice, and transported to the laboratory. The standard literature of taxonomic guides was used to identify the collected specimens (Clark & Rowe, 1971).

Echinochrome extraction

Sea urchin shells and spines were used for Ech pigment extraction by a slightly modified Amarowicz method (Amarowicz, Synowiecki & Shahidi, 1994). After evacuating the inner tissues, the shells and spines were washed with cold water, air-dried at 4 °C for 24 h in the dark, and then grounded. The powder (10 g) was dissolved by gradually adding 20 ml of 6 M HCl. The solution pigments were extracted three times with equal diethyl ether volumes. The acid was eliminated from the collected ether layer by using a 5% NaCl wash. The ether solution was dried over anhydrous sodium sulfate before the solvent was evaporated under reduced pressure. The polyhydroxylated naphthoquinone pigment extract was kept in the dark at −30 °C.

High-performance liquid chromatography analysis

A Shimadzu high-performance liquid chromatography (HPLC) system with double LC20AD pumps, a DGU-20 A3 degasser, and an SPD-M20 A diode-array detector was used. With a 1.0 mL/min flow rate, chromatographic separation was performed using a Zorbax Eclipse Plus C18 column (250 mm, 4.6 mm, 5 m, Agilent Acetonitrile/methanol (5:9) and 0.1% formic acid made up the binary mobile phase. An elution profile looked like this: 30–80% acetonitrile in formic acid for 0–25 min (linear gradient). The injection volume was 20 µL. Between 200 and 800 nm, the detection was noted. The data analysis system comprised the L.C. Solution (Shimadzu). Ech was dissolved in 5 mg/mL DMSO.

Mass spectroscopy

Shimadzu QP 2010 Plus (Kyoto, Japan) was utilized for LC-MS analysis. Electron energy was 70 eV. The mass range from 50 to 500 m/z was scanned at 1,000 amu/s, corresponding to 0.5 event time. The ion source was 250 °C.

Experimental animals

Eighteen male albino Wistar rats (Rattus norvegicus) 190 g ± 10 g were brought from the National Research Center, Cairo, Egypt. Rats were housed and divided in polyacrylic cages (three per cage) in the well-ventilated animal house of the Zoology Department, Faculty of Science, Cairo University. Free access to food and water was allowed, and animals were kept in a 12 h light-dark cycle environment at room temperature 21 °C (±3 °C) and relative humidity (45 ± 5%). Before enrolling in the experiment, rodents were kept in the laboratory for 15 days to exclude infection and adapt to laboratory conditions.

Ethical consideration

All the study experimental protocols and procedures followed international guidelines for the care and use of laboratory animals and were approved by the Institutional Animal Care and Use Committee (IACUC) of Cairo University Faculty of Science approved with the approval number: CUIF0823.

Experimental design

The animals were divided into three equal groups (n = 6): The control group: Rats were intraperitoneally (IP) injected with saline for 2 days, then administered 5% DMSO daily orally for six consecutive days.

The anemia group: Rats were IP injected with 40 mg/kg/day PHZ for 2 days (Diallo et al., 2008), then administered 5% DMSO/day orally for six consecutive days.

The Ech group: Rats were IP injected with 40 mg/kg/day PHZ for 2 days, then orally administered 1 mg/kg/day Ech (Mohamed et al., 2019) daily for six consecutive days.

Animal handling and specimen collection

After 3 days of the experiment, blood samples were withdrawn from the retro-orbital sinus in EDTA-tubes for a complete blood count (CBC). By day 6, rats were anesthetized by IP injection of 50 mg/kg body weight sodium pentobarbital (Mohamed et al., 2020). Blood samples were collected by exsanguination and divided into two tubes: one for CBC, and the second was centrifuged for 20 min at 3,000 rpm. Rat sera, from second tubes, were pipette-aspirated into sterilized Eppendorf tubes and stored at −80 °C for biochemical analysis. Following blood collection, the rats were dissected. The livers and kidneys were quickly removed, washed with saline, and divided into two pieces; one homogenized for the biochemical analyses and the other for histopathological examination.

Analysis of hematological parameters

CBC was determined using automatic counter Sysmex (K21, Tokyo, Japan).

Liver and kidney homogenate preparation

Liver and kidney tissues were homogenized (10% w/v) in ice-cold Phosphate buffer (50 mM, pH 7.4) and centrifuged at 3,000 rpm for 15 min at 4 °C. The supernatant was used for the oxidative stress analyses.

Biochemical assessment and oxidative stress markers

The serum levels of AST, ALT, ALP, gamma-glutamyl transferase (GGT), total bilirubin, creatinine, urea, and uric acid, together with liver and kidney homogenate supernatant levels of malondialdehyde (MDA), nitric oxide (NO), glutathione (GSH), catalase (CAT), glutathione-S-transferase (GST) were determined according to the manufacturer’s instructions using the Spectrum Diagnostics and Biodiagnostic kits (Giza, Egypt).

Histopathology

Liver and kidney specimens were collected from rats of all three study groups. They were immediately fixed in 10% neutral-buffered formalin. After appropriate fixation, tissue samples were dehydrated in ethyl alcohol, cleared in xylol, embedded, and cast in paraffin. Paraffin tissue sections 4 uM-thick were prepared and routinely stained with hematoxylin and eosin stains for histopathological evaluation. A lesion-score was developed to evaluate liver and kidney lesions. For liver evaluation, a total score was obtained by summating scores given for hepatocellular degeneration and inflammatory cell infiltration (ranging from 0 to 3 (0 = absent, 1 = mild, 2 = moderate, and 3 = severe)). For renal tissue evaluation, the total score was obtained by evaluating inflammatory cells infiltration and renal tubular damage severity on a scale of 0 to 3, respectively.

Statistical analysis

All values were expressed as means ± standard error of the mean (SEM). The one-way analysis of variance (ANOVA) was used for comparisons within groups, followed by the Duncan post hoc test using SPSS (SPSS Inc., Chicago, IL, USA) software. P values < 0.05 were considered statistically significant.

Results

HPLC analysis of Ech

As shown in (Fig. 1A), the HPLC analyses of separated Ech revealed a significant peak with a 5.02 min retention time, matching the standard Ech with a total purity of 87.51%.

Figure 1 (A) HPLC analysis of echinochrome, and (B) the mass spectrum of echinochrome A.

The mass spectra

There were prominent peaks at (m/z) 265 in the LC-MS spectrum of echinochrome A, which corresponds to its molecular weight (Fig. 1B).

Hematological markers

As shown in Table 1, after 3 days of the experiment, the WBCs and platelets counts increased significantly (P < 0.05), while the RBCs and Hb concentration decreased significantly (P < 0.05) in the anemia group compared to the control group. On the other hand, the WBCs and platelets counts decreased significantly (P < 0.05), and RBCs and Hb concentration increased significantly (P < 0.05) in the Ech group compared to the anemia group.

Table 1 Therapeutic effect of Ech on hematological parameters among the experimental groups after 3 and 6 days.

Experimental period	Groups	WBCs	RBCs
(million/cm)	Hb
(g/dl)	Platelets	
After 3 days	Control	12.00 ± 0.51	5.98 ± 0.10	14.68 ± 0.21	549.40 ± 17.67	
Anemia	49.67 ± 2.02*	2.82 ± 0.10*	10.58 ± 0.67*	745.67 ± 27.57*	
Ech	45.26 ± 1.05*^	3.34 ± 0.08*^	13.45 ± 0.37^	606.00 ± 24.82^	
After 6 days	Control	6.62 ± 0.23	6.49 ± 0.16	13.45 ± 0.31	605.33 ± 10.08	
Anemia	4.46 ± 0.08*	2.99 ± 0.10*	10.25 ± 0.27*	439.60 ± 10.95*	
Ech	5.68 ± 0.30*^	3.72 ± 0.07*^	11.45 ± 0.08*^	513.67 ± 23.39*^	
Notes:

Values are presented as mean ± SEM.

Values are significantly different (P < 0.05) when compared to the control group (*) and the anemia group (^).

After 6 days of the experiment, the WBCs, RBCs, and platelets counts, and Hb concentration decreased significantly (P < 0.05) in the anemia group compared to the control group, while these counts and Hb level increased significantly (P < 0.05) in the Ech group compared to the anemia group.

Serum biomarkers

Table 2 showed a significant (P < 0.05) increase in serum AST, ALT, ALP, GGT, total bilirubin, creatinine, urea, and uric acid levels in HA rat group compared to the control group. Meanwhile, oral administration of Ech significantly (P < 0.05) decreased hepatorenal markers compared to the anemia group.

Table 2 Therapeutic effect of Ech on serum biochemical parameters among the experimental groups after 3 and 6 days.

Groups	AST (U/ml)	ALT (U/ml)	ALP (U/L)	GGT (U/L)	Total bilirubin
(mg/dL)	Total protein
(g/dL)	Albumin
(g/dL)	Creat.
(mg/dl)	Urea
(mg/dl)	Urate
(mg/dl)	
Control	17.83 ± 2.37	14.00 ± 1.00	19.63 ± 2.21	1.67 ± 0.21	0.50 ± 0.01	4.96 ± 0.34	3.26 ± 0.10	1.61 ± 0.12	28.66 ± 0.68	7.48 ± 0.18	
Anemia	46.83 ± 3.40*	25.33 ± 1.12*	38.03 ± 3.40*	5.19 ± 0.64 *	1.21 ± 0.05*	4.62 ± 0.27	3.14 ± 0.03	2.33 ± 0.05*	45.23 ± 1.98*	14.23 ± 1.00*	
Ech	27.67 ± 1.50*^	17.67 ± 0.61*^	28.28 ± 2.53*^	2.45 ± 0.44^	0.84 ± 0.01*^	4.63 ± 0.43	3.13 ± 0.06	1.66 ± 0.12^	34.57 ± 0.86*^	9.92 ± 0.67*^	
Notes:

Values are presented as mean ± SEM.

Values are significantly different (P < 0.05) when compared to the control group (*) and the anemia group (^).

Oxidative stress markers

A significant (P < 0.05) increase in the MDA and NO concentrations in the anemia group was observed, while GSH, CAT, and GST levels were reduced compared to the control group. Following Ech treatment, Rats showed significantly (P < 0.05) decreased MDA and NO concentrations, with increased GSH, CAT, and GST levels compared to the anemia groups, as demonstrated in Table 3.

Table 3 Therapeutic effect of Ech on liver oxidative stress parameters among the experimental groups after 3 and 6 days.

Groups	Organ	MDA
(nmol)	GSH
(mg)	CAT
(U)	NO
(μM)	GST
(μM/min)	
Per gm tissue	
Control	Liver	0.63 ± 0.02	4.78 ± 0.71	198.61 ± 6.07	213.52 ± 21.23	1.62 ± 0.17	
Kidney	1.67 ± 0.21	3.42 ± 0.07	919.02 ± 101.90	180.34 ± 17.67	1.14 ± 0.16	
Anemia	Liver	0.83 ± 0.04*	1.61 ± 0.13*	128.01 ± 4.31*	999.10 ± 38.41*	0.54 ± 0.03*	
Kidney	3.52 ± 0.26*	1.80 ± 0.07*	291.58 ± 16.25*	756.03 ± 23.27*	0.36 ± 0.07*	
Ech	Liver	0.51 ± 0.03*^	3.09 ± 0.10*^	170.22 ± 4.87*^	574.11 ± 50.05*^	0.93 ± 0.08*^	
Kidney	1.55 ± 0.05^	2.67 ± 0.09*^	521.59 ± 9.14*^	307.05 ± 17.33^	0.87 ± 0.08*^	
Notes:

Values are presented as mean ± SEM.

Values are significantly different (P < 0.05) when compared to the control group (*) and the anemia group (^).

Histopathology results

The control group showed normal hepatic lobules with normally-positioned central veins and radiating cords of intact hepatocytes. Each hepatocyte revealed abundant cytoplasm with a central nucleus. The anemic group showed frequent portal inflammation, as illustrated by the inflammatory cell aggregation in the portal triads, with a statistically significant higher hepatic lesion scoring compared to the control group. Degeneration of hepatocytes was observed along with disorganization of hepatic cords and accumulation of golden yellow to brown hemosiderin pigment in the hepatic sinusoids and the cytoplasm of affected hepatocytes. Marked improvement was observed in the Ech group with apparently normal hepatic parenchyma in multiple sections examined with fewer ones showing hemosiderin pigments accumulation, with a statistically significant lower hepatic lesion scoring compared to the anemic group, as shown in Fig. 2.

Figure 2 Photomicrographs of the liver in different experimental groups (H&E 400X).

(A) Control group showing normal histology of hepatic cords. (B) Anemia group showing variable of portal inflammatory cell infiltration (red arrow). (C) Ech group shows less accumulation of hemosiderin pigment in the hepatic sinusoids (arrow). Values are significantly different (P < 0.05) when compared to the control group (*) and the anemia group (^).

Examination of the renal tissue from the control group revealed unremarkable histologic architecture of glomeruli and renal tubules with intact renal cortical and medullary basement membranes. The anemia group showed numerous histopathologic changes including multifocal interstitial nephritis and periglomerular inflammatory cell infiltrate, degeneration and necrosis of renal tubular epithelium in several sections examined, and eosinophilic renal tubular cast accumulation, with a statistically significant higher renal lesion scoring compared to the control group. Marked improvement was detected in the Ech group that revealed normal renal parenchyma with less inflammatory cell aggregates, with a statistically significant lower renal lesion scoring compared to the anemic group, as shown in Fig. 3.

Figure 3 Photomicrographs of kidneys in different experimental groups (H&E, 400X).

(A) Control group showing normal histology of the renal cortex. (B) Anemia group showing numerous interstitial and periglomerular inflammatory cell infiltration (arrow). (C) Ech group shows fewer inflammatory cells infiltrating the renal cortex (arrow). E Values are significantly different (P < 0.05) when compared to the control group (*) and the anemia group (^).

Discussion

Hemolytic anemia (HA) has multiple underlying causes and consequences. The current study has revealed a significant decrease in Hb concentration and RBC count after PHZ injection. PHZ interacts with the plasma membrane of RBCs and produces reactive oxygen species (ROS), lipid peroxidation, hemoglobin oxidation and precipitation, and eventually premature hemolysis with a shorter lifespan (Pandey et al., 2014). The elevated Hb concentration and RBCs count to about normal levels following Ech treatment may indicate a possible antioxidant protective effect. Ech pigment has a powerful anti-inflammatory effect due to a group of bioactive polyketide compounds (Ghelani et al., 2022) might induce cholinergic activation with anti-inflammatory effects (Sadek et al., 2022). In addition, Ech can protect RBC membranes from lysis by stabilizing them better than the standard acetylsalicylic acid drug (Ghelani et al., 2022). In the current study, After 3 days of treatment, Ech group Hb content was better than after 6 days. Regenerative erythropoiesis and increased erthroptioein levels did not begin in rats within 4 days of anemia induction (Chen, Feng & Jeng, 2018; Bamba et al., 1999). Ech’s ability to stimulate erythropoiesis at this time point is clearly responsible for the restoration of Hb levels. Ech A was found to be an effective agent for promoting cell proliferation and maintaining the stemness of hematopoietic stem and progenitor cells (Park et al., 2019). However, the improvement in Hb level due to Ech weaks the stimulus (oxygen content) for erythropoetin secretion on day 6.

The white blood cells (WBCs), also known as leukocytes, are the backbone of the innate and acquired immune systems protecting human body against harmful pathogens (Diallo et al., 2008). The current study has revealed a significant increase in the WBC counts after 3 days of PHZ injection. This leukocytosis observed with blood loss or induced by PHZ administration was linked to a stress reaction (Criswell et al., 2000). PHZ injection for 6 days causes a significant decrease in the WBCs count which was though by previous researchers to represent a recovery from PHZ toxicity, decreasing the WBCs count (Sonoda & Sasaki, 2012). In our study, the WBCs count was significantly reduced after Ech treatment for 3 days, while 6 days of treatment caused a significant increase in the WBCs count. The mechanism of this WBC count alteration is not clear and may involve reticuloendothelial system sequestration or bone marrow effect. Ech has been shown to contributes to immune system alternation by controling regulatory T cells generation and inhibiting pro-inflammatory cytokines IL-1β,IL-6, IL-8, TNF-α, INF-α, and NKT production, thus restoring the immune balance (Rubilar et al., 2021).

Platelets mediate blood clotting at vascular damage sites maintaining circulatory hemostasis. They also play a role in inflammation, angiogenesis, and innate immunity, among other non-hemostatic processes (Sun, Zhou & Ye, 2021). The present study has revealed a significant increase in platelets count 3 days after PHZ injection. This may happen due to acute infection, inflammation, or chronic inflammatory diseases such as PHZ-induced anemia (Evstatiev et al., 2014). On the other hand, PHZ injection for 6 days significantly reduced platelet count. The decrease in platelets count is related to decreased inflammatory response following recovery from PHZ toxicity (Sonoda & Sasaki, 2012). Ech enhances regulatory T cell generation altering inflammatory and immune responses. Nevertheless, Ech induces production of macrophage type-M2, promoting inflammation resolution and initiating tissue repair (Oh et al., 2019).

The study showed a PHZ-induced marked elevation in hepatic enzymes and bilirubin levels. exposure to chemicals, drugs, or virus infections induce acute liver injury (Ramirez et al., 2018) Due ot its effect on elevating liver enzymes, PHZ has been proposed as to have a hepatotoxic effect on hepatocytes (Obayuwana et al., 2022) Chronic liver disease result in normocytic and macrocytic anemias (Allahmoradi, Alimohammadi & Cheraghi, 2020). PHZ induces jaundice through direct oxidation of Hb o generating oxyhemoglobin and methemoglobin, and RBC hemolysis increasing levels of unconjugated bilirubin (Nawaz, Shad & Iqbal, 2016; Zhang et al., 2015). Levels of total bilirubin increase due to decreased uptake and conjugation by the dysfunctioning hepatocytes. The reduction in AST, ALT, ALP, GGT, and total bilirubin to near-normal levels seen in rat group treated by Ech, suggests an indirect protective effect on hepatocytes (Soliman, Mohamed & Marie, 2016). Also, the antioxidant properties of Ech helps hepatocyte recovery from injury by promoting antioxidative and detoxification mechanisms (Mohamed et al., 2019).

Creatinine, urea, and urate concentrations are useful serum markers for glomerular filtration and renal functions (Dakrory et al., 2015). In this study, we noticed elevated renal-function markers following PHZ intoxication which was previously attributed to by alternations in mRNA levels of genes responsible for renal tubular damage and vascular inflammation in rats exposed to PHZ (Merle et al., 2018). Moreover, an PHZ-induced increase in hemolysis with further renal tubular injury was evident in the renal pathogenesis (Parvaz et al., 2022). In this study, our findings support what have been previously shown that treatment with Ech results in reduced serum creatinine, urea, and uric acid concentrations, and restores renal histophysiologic features (Fahmy et al., 2019).

Lipid peroxidation is one of many consequence of excess ROS production at celluar level, with subsequent malondialdehyde (MDA) generation. The endothelial nitric oxide synthase (eNOS) enzyme generates the biological mediator “NO” (Albrecht et al., 2003) the latter of which plays a role in reducing cellular oxidative stress (Rojo et al., 2014). PHZ is a potent oxidizing agent that generates a wide array of PHZ-derived radicals such as phenyldiazene, phenylhydrazyl radical, and benzene-diazonium ions, and our study showed elevated serum MDA and NO levels in the PHZ-exposed rats (Parvaz et al., 2022). Our findings showed that treating anemic rats with Ech significantly decreased MDA and NO levels which posssbily can be explained by scavenging active oxygen radicals (Nawaz, Shad & Iqbal, 2016) or inhibiting lipid peroxidation (Lebedev, Ivanova & Levitsky, 2008). As an antioxidant, Ech can bind to ROS deactivating them and protecting against oxidative stress and cellular damage.

Generally, antioxidants perform their function through enzymatic and non-enzymatic mechanisms (Kapoor & Gupta, 2020). Enzymatic antioxidants such as SOD, CAT, and glutathione-S-transferase (GST) act by eliminating free radicals or breaking them down. Non-enzymatic antioxidants, on the other hand such as vitamin C, vitamin E, carotenoids, and glutathione (GSH), act by interrupting free-radical-chain reactions (Qin, 2018). The current work showed decreased GSH content and CAT and GST activities following PHZ injection. GSH, CAT, and superoxide are naturally occurring compounds in erythrocytes, contributing to the antioxidant capacity of the blood, and reducing their cellular levels, as seen in anemia, results in oxidative stress and, subsequently, HA (Madhikarmi & Murthy, 2015). The present study also showed markedly suppressed PHZ-induced lipid peroxidation and enhanced antioxidant activity (GSH, GST, and CAT) in hepatorenal tissues. indicating possible ROS scavenging activity.

Conclusion

Ech had some protective effect against biopathologic features see in phenylhydrazine-induced hemolytic anemia in rats, possiblydue to its anti-inflammatory and antioxidant properties.

Supplemental Information

Supplemental Information 1 Biomarkers.

Click here for additional data file.

Supplemental Information 2 Oxidative stress.

Click here for additional data file.

Supplemental Information 3 CBC.

Click here for additional data file.

Supplemental Information 4 P-values.

Click here for additional data file.

Supplemental Information 5 ARRIVE 2.0 Checklist.

Click here for additional data file.

The authors thank all the academic and technical staff that provided administrative and technical support.

Additional Information and Declarations

Competing Interests

Author Contributions

Animal Ethics

Data Availability

The authors declare that they have no competing interests.

Mona S. E. F. El-Shehry conceived and designed the experiments, performed the experiments, prepared figures and/or tables, and approved the final draft.

Rafa A. Amrymi conceived and designed the experiments, performed the experiments, prepared figures and/or tables, and approved the final draft.

Tarek Atia performed the experiments, analyzed the data, prepared figures and/or tables, authored or reviewed drafts of the article, and approved the final draft.

Bassant M. M. Lotfy conceived and designed the experiments, performed the experiments, prepared figures and/or tables, and approved the final draft.

Salma H. A. Ahmed conceived and designed the experiments, performed the experiments, prepared figures and/or tables, and approved the final draft.

Sarah A. Qutb conceived and designed the experiments, performed the experiments, prepared figures and/or tables, and approved the final draft.

Sara B. Ali conceived and designed the experiments, performed the experiments, prepared figures and/or tables, and approved the final draft.

Ayman S. Mohamed conceived and designed the experiments, performed the experiments, analyzed the data, prepared figures and/or tables, authored or reviewed drafts of the article, and approved the final draft.

Mohamed R. Mousa conceived and designed the experiments, performed the experiments, prepared figures and/or tables, and approved the final draft.

Ahmed A. Damanhory performed the experiments, analyzed the data, prepared figures and/or tables, authored or reviewed drafts of the article, and approved the final draft.

Mostafa E. Metawee performed the experiments, analyzed the data, prepared figures and/or tables, authored or reviewed drafts of the article, and approved the final draft.

Hader I. Sakr performed the experiments, analyzed the data, prepared figures and/or tables, authored or reviewed drafts of the article, and approved the final draft.

The following information was supplied relating to ethical approvals (i.e., approving body and any reference numbers):

The Cairo University, Faculty of Science Institutional Animal Care and Use Committee (IACUC) approved this study’s experimental protocols and procedures with the approval number: CUIF0823.

The following information was supplied regarding data availability:

The data is available at figshare: sakr, hader (2023). Hematopoietic effect of Echinochrome on phenylhydrazine-induced hemolytic anaemia in rats. figshare. Dataset. https://doi.org/10.6084/m9.figshare.22557805.v1.

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
