# Peer review of "Hematopoietic effect of echinochrome on phenylhydrazine-induced hemolytic anemia in rats"

_PeerJ, doi:10.7717/peerj.16576_

## Round 0.1 · original submission · Major Revisions

I have completed my evaluation of your manuscript. The reviewers recommend reconsideration of your manuscript following major revision. I invite you to resubmit your manuscript after addressing the comments below. When revising your manuscript, please consider all issues mentioned in the reviewers' comments carefully: please outline every change made in response to their comments and provide suitable rebuttals for any comments not addressed. Please note that your revised submission may need to be re-reviewed.

·

Basic reporting

used for histological studies. The microscopic images have been presented satisfactorily. Quantification of nuclear area is highly recommended. Authors are requested to provide single image of each group, bearing all important identification factors. Since it is not in much higher magnification, providing a single field is more scientific.

These all issues may bring more firmness to the study. Sentence making errors must be avoided e.g.; first line of Section 13 of 'Method' section. Again, to mention, hematopoietic effect has not been proven here.

Experimental design

Seems all right.

Validity of the findings

The findings seems valid.

Additional comments

The paper needs major revision.

Reviewer 2 ·

Basic reporting

In this manuscript, the authors studied the therapeutic effect of Ech on phenylhydrazine (PHZ )-induced H.A. in rats. The experiments were designed very well and carried out the study and provided supportive data for this study. Histopathological investigation of the liver and kidney showed improvement. The treatment of Ech in the PHZ induced rats, showed the hepatorenal protective effect. Despite that, the following corrections to be made before considering for the publication.

Experimental design

No comment

Validity of the findings

No comment

Additional comments

It’s not clear that how was the platelets were counted, please mention in the manuscript.
Animals were injected 40 mg/kg of Ech. Maximum tolerance dose was tested before?
Page 9, line 129, 40mg/kg/day should be 40 mg/kg/day.
Page 9, line 131, 40mg/kg/day should be 40 mg/kg/day.
Page 9, line 132, 1mg/kg/day should be 1 mg/kg/day.
Page 9, line 141, “The sera were kept at -80°C, pending biochemical analysis”, I assume it is “The sera were kept at -80°C, for pending biochemical analysis”.
Page 10, line 148, “50mM”. it should be 50 mM.
Page 10, line 149, “15 min. at four °C.” it should be 15 min at four °C.
Page 10, line 169, “concentration of 87.51%” I assume it is “purity of 87.51 %”
Page 13, line 252, it’s not very clear from the statement “The declined AST, ALT, ALP, GGT, and total bilirubin to about the normal levels by the treatment with Ech, indicating maintained function and structure of hepatocytes”. Please correct the sentence.
Page 22, table 2, table legends should be included for a, b, c.
Page 25, table 3, table legends should be included for a, b, c.

---

## Round 0.2 · Major Revisions

I have completed my evaluation of your manuscript. The reviewers recommend reconsideration of your manuscript following major revision. I invite you to resubmit your manuscript after addressing the comments below. When revising your manuscript, please consider all issues mentioned in the reviewers' comments carefully: please outline every change made in response to their comments and provide suitable rebuttals for any comments not addressed. Please note that your revised submission may need to be re-reviewed.

**Language Note:** The review process has identified that the English language must be improved. PeerJ can provide language editing services - please contact us at [email protected] for pricing (be sure to provide your manuscript number and title). Alternatively, you should make your own arrangements to improve the language quality and provide details in your response letter. – PeerJ Staff

·

Basic reporting

The present manuscript entitled “Hematopoietic effect of Echinochrome on phenylhydrazine-induced hemolytic anemia in rats” in its revised version has presented the protective role of echinochrome toward hepatic and renal tissues of anemic rats which may project Ech as a therapeutic for anemic patients.

This reviewer thinks that, this research is bringing light to a new regimen of echinochrome, since no such studies have been reported earlier, though antioxidative effects are well established now.

The reviewer would like to suggest the authors to mention some specific names of chemicals/drugs, obtained from Ech and their utility in present medical scenario.

In abstract and in ‘Conclusion’ section as well, authors have mentioned about anti-inflammatory role of Ech. However, they have not checked about any pro and/or anti-inflammatory markers and hence it can’t be concluded from this study. The reviewer strongly suggests measurement of some of these markers.

Mention the magnification used in capturing HE stained tissue images.

In conclusion, authors should mention the sole findings of their own work. There is no need to present any other works as reference. Authors have not checked about oxygen free radical scavenging ability of Ech also in their study. Mention only if it is performed. The reviewer strongly suggests though.

The Hb content in Ech group has shown better result in 3days treatment compared to 6 days treatment. This finding must be discussed in one or two lines.

Some sentence making and typographical errors are there throughout the MS, which should be taken care of in next version.

Experimental design

Seems alright

Validity of the findings

Good findings but the manuscript needs revision further

Additional comments

REVISION NEEDED

Reviewer 2 ·

Basic reporting

No Comments

Experimental design

No Comments

Validity of the findings

No Comments

Additional comments

The manuscript should undergo serious English corrections. The authors may seek English editing service to correct the manuscript. The authors were failed to satisfy the reviewers comments.
Lot of extra space between the words (numerous places), and spelling mistakes, grammatical errors.
These are few examples for the mistakes.
Page 2, line 13, “hemoglobin, , reduced”
Page 2, Line 26, “light headedness”
Page 3, line 21, “have shown to have some”
Page 3, Line 23, “is one of multiple spinochrome”
Page 5, line 27, “from second tubes”
Page 6, line 17, “uM-thin”
Page 10, line 5, “alternations,”
Page 11, Line 7, “an PHZ-induced,”
Page 11, Line 8, “In this study, our findings support what have been previously shown that”
Page 11, Line 11, “one of many consequence”
Line 15, “that revealed normal”

Page 4, line 24, “An elution profile looked like this: 30-80% acetonitrile”- authors should use scientific written terms to write the manuscript.

Authors can discuss more about the antioxidant, antiviral, and antimicrobial activity of Echinochrome and provide more evidence.
How did the authors confirm that the extraction of Echinochrome from the Sea urchins. I assume that we cannot assure that the extracted compound is Echinochrome, only by HPLC, in addition we need ES-MS

---

## Round 0.3 · Minor Revisions

I have completed my evaluation of your manuscript and I found authors have addressed all the concerns raised in the previous version of the manuscript and the quality has improved after incorporating the required modifications. Therefore, the manuscript may be considered for publication in this Journal.

Before a final acceptance, merging figures 1 and 2 into one figure and assembling figures 4, 5, and 6 into a single figure would improve the readability of this manuscript.

---

## Round 0.4 · accepted · Accept

It is a pleasure to accept your manuscript entitled " Hematopoietic effect of Echinochrome on phenylhydrazine-induced hemolytic anaemia in rats" in its current form for publication in PeerJ.